# Intense Locomotion Enhances Oviposition in the Freshwater Mollusc *Lymnaea stagnalis*: Cellular and Molecular Correlates

**DOI:** 10.3390/biology12060764

**Published:** 2023-05-24

**Authors:** Ilya Chistopolsky, Alexandra Leonova, Maxim Mezheritskiy, Dmitri Boguslavsky, Angelina Kristinina, Igor Zakharov, Andrey Sorminskiy, Dmitri Vorontsov, Varvara Dyakonova

**Affiliations:** Koltzov Institute of Developmental Biology of the Russian Academy of Sciences, 119334 Moscow, Russia; ilyachistopolskiy@idbras.ru (I.C.); alex.leonova.a@gmail.com (A.L.); m.mezheritskiy@idbras.ru (M.M.); boguslavsky@idbras.ru (D.B.); aa.kristinina@idbras.ru (A.K.); iszakharov@idbras.ru (I.Z.); andy14264@gmail.com (A.S.); d.vorontsov@idbras.ru (D.V.)

**Keywords:** exercise, locomotion, physical activity, adaptation to environmental change, reproduction, neuronal mechanisms of reproductive activity, serotonin, oviposition, ovulation hormone, asymmetry

## Abstract

**Simple Summary:**

Whether reproductive behaviour is affected by physical activity (exercise) remains largely unknown. We addressed this question using a simple model organism, the pond snail *Lymnaea stagnalis*. In this animal, one can investigate events at the molecular and cellular levels. We found that intense crawling in shallow water (exercise) for two hours resulted in an increased number of egg clutches and the total number of eggs laid in the following 24 h. This effect was stronger from January to May, in contrast to September–December. Synthesis of RNA transcripts of the egg-laying hormone was higher in the central nervous system of snails that exercised. Additionally, the CDC neurons, which produce the egg-laying hormone and play a key role in egg-lying behaviour, showed a stronger response to electrical stimulation in exercised snails. Our data suggest that exercise enhances reproduction in *L. stagnalis*, in spite of its obvious energetic costs. The finding agrees with the hypothesis that behavioural effects of exercise were formed early in evolution and have an adaptive significance: to prepare an organism to enter a new environment.

**Abstract:**

Intense species-specific locomotion changes the behavioural and cognitive states of various vertebrates and invertebrates. However, whether and how reproductive behaviour is affected by previous increased motor activity remains largely unknown. We addressed this question using a model organism, the pond snail *Lymnaea stagnalis*. Intense crawling in shallow water for two hours had previously been shown to affect orienting behaviour in a new environment as well as the state of the serotonergic system in *L. stagnalis*. We found that the same behaviour resulted in an increased number of egg clutches and the total number of eggs laid in the following 24 h. However, the number of eggs per clutch was not affected. This effect was significantly stronger from January to May, in contrast to the September–December period. Transcripts of the egg-laying prohormone gene and the tryptophan hydroxylase gene, which codes for the rate-limiting enzyme in serotonin synthesis, were significantly higher in the central nervous system of snails that rested in clean water for two hours after intense crawling. Additionally, the neurons of the left (but not the right) caudo-dorsal cluster (CDC), which produce the ovulation hormone and play a key role in oviposition, responded to stimulation with a higher number of spikes, although there were no differences in their resting membrane potentials. We speculate that the left–right asymmetry of the response was due to the asymmetric (right) location of the male reproductive neurons having an antagonistic influence on the female hormonal system in the hermaphrodite mollusc. Serotonin, which is known to enhance oviposition in *L. stagnalis*, had no direct effect on the membrane potential or electrical activity of CDC neurons. Our data suggest that (i) two-hour crawling in shallow water enhances oviposition in *L. stagnalis*, (ii) the effect depends on the season, and (iii) the underlying mechanisms may include increased excitability of the CDC neurons and increased expression of the egg-laying prohormone gene.

## 1. Introduction

Behavioural modulation of reproductive functions is an important field in modern medicine and developmental biology. In humans, the influence of physical activity on male [1] and female [2] fertility, embryonic gestation [3] and offspring health [4,5,6] is being actively studied. However, little is known about the mechanisms underlying this influence and the possible adaptive significance.

We previously suggested that behavioural effects of exercise (or intense locomotion) were formed on an ancient evolutionary basis and have an adaptive significance: to prepare an organism to enter a new environment [7,8,9,10,11]. The latter is characterized by higher uncertainty; therefore, the biological mechanisms of preadaptation to novelty are likely to include cognitive and reproductive activation. The former helps to adapt the behaviour of the organism itself, allowing it to survive in less familiar environments, while the latter may contribute to the survival of its ”genetic material” by increasing the number of descendants and their diversity. Activation of cognitive functions through exercise has already been noted in an impressive number of studies on different groups of animals (for a review, see [11,12,13]), including humans [13] and various model invertebrates [10]. The impact of intense locomotion on reproductive behavior in model organisms has not been studied to a similar extent.

This study was carried out to test the hypothesis that intense locomotion, namely crawling in shallow water, will enhance reproductive activity (oviposition) in a novel environment in the pond snail *Lymnaea stagnalis*. This snail is a model system for studying cellular and molecular neurobiology, developmental biology and evolutionary biology [14,15,16,17,18,19]. Previously, we demonstrated that two hours of crawling in shallow water affected decision-making in a novel environment [7] as well as the biophysical properties of serotoninergic neurons associated with locomotion [20] and the metabolism of serotonin [8]. Interestingly, activation of serotonin synthesis has been shown to increase fertility in *L. stagnalis* [21]. These data suggesting an activation of the serotonin system by intense crawling on the one hand and stimulating effect of serotonin on oviposition on the other were an additional rationale for our hypothesis on the potentiating effect of intense locomotion on reproductive behaviour.

In *L. stagnalis*, egg-laying is controlled by the egg-laying hormone produced by caudo-dorsal cerebral (CDC) neurons [22]. This hormone is called the caudo-dorsal cell hormone (CDCH) or ovulation hormone [23,24,25]. The transcript encoding this product has been sequenced and is called ovulation prohormone or CDCH-1 [23]. Several peptides arise from the ovulation prohormone gene, of which the ovulation hormone is the best investigated and understood. This peptide is released into the haemolymph and evokes ovulation, mass egg formation and egg-laying behaviour [26].

Caudodorsal neurons that produce the ovulation hormone form a pair of clusters that are located in the right and left cerebral ganglia near the cerebral commissure, 30 cells in the left ganglion and 70 cells in the right ganglion [24,27]. Prior to egg-laying, CDC neurons switch to intense electrical activity, which promotes the simultaneous release of ovulation prohormone [27,28]. Their synchronous activity is based on electrical coupling and on the chemical positive feedback, namely the excitatory effect of CDCH on their electrical activity [27,29,30]. After eggs are laid, the CDCs are inhibited for several hours and then return to the resting state; they may switch to the active state again [28].

The available data provide useful background for testing the above hypothesis on the stimulating effect of exercise on reproduction at the behavioural, cellular and molecular levels in *L. stagnalis*. In this study, we analysed the number of clutches and eggs laid after intense crawling, the level of transcripts of ovulation prohormone and tryptophan hydroxylase genes in the central ganglia and several biophysical properties of CDC neurons. The obtained results suggest that two hours of exercise can indeed enhance oviposition in *L. stagnalis*.

## 2. Materials and Methods

### 2.1. Study Design

Three approaches were used to test the hypothesis. The first was a behavioural study, in which the numbers of clades and eggs were evaluated 24 h after intense crawling. In the second approach, RT-PCR was used to estimate possible changes in the expression of a key gene, ovulation prohormone, that is involved in oviposition and to evaluate the response of the serotonin synthesis rate-limiting gene, tryptophan hydroxylase, to intense crawling. Third, using electrophysiological experiments with microelectrode intracellular recordings, we tested the possible changes in the biophysical properties of CDC neurons secreting ovulation hormone after intense crawling. In the latter case, we also tested whether serotonin can activate the CDC neurons, providing a direct link between locomotion and oviposition. Randomized controlled trials were used in all experiments.

### 2.2. Animals

*Lymnaea stagnalis* snails were taken from a breeding colony initially obtained from the Netherlands (1992) and then bred multiple times (1995–2019) with specimens from the wild population of the Oka River, Moscow region, Russia. Snails were kept in aquariums (100 L) containing permanently aerated water, with no filtering or artificial water flow, at a temperature of 23–25 °C, under natural light conditions and fed on lettuce ad libitum. Every two weeks, the water was changed. Sexually mature snails with a shell length of 2–3 cm were taken for experimentation during periods between water changes. The snails were randomly assigned to experimental and control groups.

### 2.3. Procedure for Crawling in Shallow Water

Snails were placed in the arena (1 m × 1 m) containing a 1-cm-deep water layer (exercised snails, as in [7,8,20]). Under these conditions, more intense terrestrial locomotion is required. In parallel, animals of the same size and age were removed from the colony and placed in plastic containers filled with water from the colony aquarium where they could drift normally (control snails). Two hours later, both experimental and control animals were placed in containers filled with clean settled water.

### 2.4. Analysis of Oviposition

In this part of the study, each experiment (29 in total) consisted of one experimental group (snails crawling in shallow water) and one control group with 8 animals in each. 24 h after crawling in shallow water, the number of egg clutches, their size (the number of eggs in a clutch), and the total number of eggs laid in the exercised and control groups were assessed. The experiments were performed from September to May, with the following number of instances in each month: Sept (6), Oct (3), Nov (3), Dec (2), Jan (7), Feb (2), Mar (0), Apr (4), May (2).

### 2.5. RT-PCR Assessment of the Expression Level of Ovulation Prohormone and Tryptophan Hydroxylase Genes

#### 2.5.1. Total RNA Extraction

Two hours after exercising the snails, the experimental and control animals were dissected in 0.1 M MgCl_2_ and the central ganglia isolated for RNA extraction and cDNA amplification. Total RNA was extracted from the homogenized tissue using the ExtractRNA kit (Evrogen, Moskva, Russia). RNA precipitate was transferred to a spin column with CleanRNA Standard (Evrogen, Moskva, Russia) where the RNA was washed and subjected to in-column DNAse I treatment as per the manufacturer’s instructions.

Total RNA was quantified using an Agilent 2100 Bioanalyzer (from Georgia Genomics and Bioinformatics Core, Athens, GA, USA; RIN value greater than seven). Total RNA samples were measured using a NanoDrop 2000 spectrophotometer (Thermo Fisher Scientific, Waltham, MA, USA) to confirm a 260:280 ratio within 2.0.

#### 2.5.2. Reverse Transcription PCR

Reverse transcription PCR was performed using the MMLV RT kit (Evrogen, Russia), which contains a mixture of oligo (dT) primers and random hexamers. Sequences of the ovulation prohormone [23] and tryptophan hydroxylase [31] transcripts were used to design primers for real-time PCR. Primers were selected using the Primer-Blast software (https://www.ncbi.nlm.nih.gov/tools/primer-blast, accessed on 1 April 2023). Oligonucleotides were synthesized by Evrogen (Moskva, Russia) for four genes; the first two were used as reference “household genes” according to [32]:

EF1a:

For [ACCACAACTGGCCACTTGATC] Rev [CCATCTCTTGGGCCTCTTTCT]

2.GAPDH:

For [CAACAACCGACAAAGCAA] Rev [CATAACAAACATAGGGGCA]

3.Ovulation prohormone *L. stagnalis*:

For [TTGTTGGTGTTCCTCCTCGAC] Rev [GGCCGTGAGCCTCGTTTTT]

4.Tryptophan hydroxylase:

For [AACCGCCCAGATAAGGTGTG] Rev [ATGGCGGAGAGTTGCGATAG]

Primer pairs were tested for formation of dimeric fragments and a search for the optimal annealing temperature was performed using gradient PCR in the range of 55–65 °C. Real-time PCR was performed in a 96-well StepOnePlus™ Applied Biosystems amplifier (Thermo Fisher Scientific, Waltham, MA, USA) using reagents from Syntol (Moscow, Russia): a set of reagents for real-time PCR in the presence of SYBR Green I dye and ROX reference dye for StepOne. The kit consists of 2.5× PCR buffer, MgCl_2_, Taq DNA polymerase with enzyme-inhibiting antibodies and deionized water.

RT-PCR experiments were conducted in October.

### 2.6. Electrophysiology

The central ganglia from a pair of snails (one exercised and one control) were isolated in 0.1 M MgCl_2_ immediately after 2 h of crawling or drifting.

The central ganglia were placed in a 2.5 mg/mL solution of pronase E (Sigma, St Louis, MA, USA) for 15 min, washed in a standard saline for snails (50 mM NaCl, 1.6 mM KCl, 4 mM CaCl_2_, 8 mM MgCl_2_, 10 mM Tris (pH 7.6)) and pinned to a 4 cm Sylgard-bottomed chamber. The central nervous systems (CNS) of each pair of snails (exercised and control) were simultaneously pinned in one experiment, in a single chamber, with a distance of approximately 1 cm between the CNS preparations. The connective tissue sheath was then removed from the cerebral ganglia. The preparations were washed under a continuous stream of saline (0.75 mL/min).

Visual identification of the CDC neurons was performed based on their location, size and coloration. The neuron was impaled with a standard glass microelectrode (10–20 MΩ filled with 3 M KCl). A standard setup for microelectrode recording was used. The electrophysiological recordings were analysed using custom software.

The following characteristics of CDC neurons were measured in the left and right symmetrical clusters: (a) spiking activity during membrane puncture by the electrode, (b) resting membrane potential (MP) and (c) the number of spikes during injection of 1 nA current for 10 s. The neuron was stimulated several times at intervals of 10–14 min. The average value was analysed (see below).

A 0.01 mM concentration of serotonin (Sigma-Aldrich, St Louis, MA, USA) was added to the continuous stream for 5 min.

All electrophysiological experiments were conducted in May and June.

### 2.7. Statistical Analysis

We used the free software PAST (“PAST: paleontological statistics software package for education and data analysis version 2.09.”, 2001, University of Oslo, Oslo, Norway) for analysis and figure preparation [33].

#### 2.7.1. Behavioural Experiments

Each experiment had a pair of control and experimental groups. Pairwise tests—pairwise *t*-test, Wilcoxon test, Sign test, Monte Carlo and Exact test—were used to analyse differences in the numbers of clutches and eggs. We treated the data as independent using Shapiro–Wilk and Anderson–Darling tests to test for normality first, followed by the *t*-test. We did not use non-parametric tests, as all variables fit the normal distribution.

Shapiro–Wilk and Anderson–Darling tests for normality followed by the *t*-test were used to assess possible differences in the sizes of clutches.

Fisher’s exact test was used to evaluate season-dependent differences in the effect of exercise on oviposition. The percentages of experiments with higher numbers of clades during the September–December and the January–May periods were compared.

#### 2.7.2. RT-PCR Data

Real-time PCR results were processed using DataAssist™ software. The program is designed for fast analysis and interactive visualization of real-time TaqMan^®^ PCR data. It uses the comparative CT method (also known as the 2^−ΔΔCt^ method) to calculate the relative values (RQ) of gene expression when comparing samples. The software uses Grubbs’ advanced outlier test, provides a measure of control gene stability based on the geNorm algorithm, and allows multiple control genes to be used for data normalization. It was used for comparison of data from control animals and animals that had undergone terrestrial locomotion and 2 h of rest (4 individuals in each group, 3 replicates).

#### 2.7.3. Electrophysiological Data

Since we had a pair of control and experimental preparations placed in the same dish and treated simultaneously in each experiment, pairwise Wilcoxon test was used to evaluate differences in activity, membrane potential and number of spikes in response to current injection between the experimental and control preparations. If in one preparation more than one neuron was investigated, the average value for the preparation was used. In addition, the data were evaluated for equal medians, Chi2, using tests for independent groups.

## 3. Results

### 3.1. L. stagnalis Oviposits More Clutches and Eggs after Crawling in Shallow Water and the Influence Is Season-Dependent

Clean water is known to stimulate egg laying in *L. stagnalis* [24,34]. Indeed, we did not observe clutches with eggs in control and exercised snails in only 6 of the 29 experiments. Previous crawling in shallow water significantly enhanced the effect of water exchange. The number of clutches laid (Figure 1A) and the total number of eggs (Figure 1B) were significantly higher in exercised snails. However, the size of individual clutches did not differ significantly between the exercised and control groups (Figure 1C).

Although we did not initially plan to test the season dependence of the crawling effect on oviposition, we found significant differences between the results of the experiments performed at different times of the year (namely, from September to December (*n* = 14) and from January to May (*n* = 15)). These periods are characterized by decreasing and increasing daylight hours, respectively. The influence of previous exercise on oviposition was significantly stronger from January to May: in the September–December period, the differences did not reach statistical significance, while in the January–May period, the effect was substantial. The effects of exercise on the number of clutches in different seasons are illustrated in Figure 2. The number of eggs is presented in Figure 3. Experiments in which a higher number of clutches were obtained in the exercised group were significantly more frequent in the January–May period (Figure 2).

### 3.2. RT-PCR Reveals a Higher Number of Transcripts of Egg-Laying Prohormone and Tryptophan Hydroxylase Genes

The above behavioural finding suggested that exercise potentiates the stimulating effect of clean water on oviposition. To see whether exercise can influence the expression of the ovulation prohormone gene, we amplified cDNA using the classical RT-PCR method.

The data obtained from control animals and animals taken 2 h after exercise (four individuals in each group, three replicates) revealed an increase in the expression level of the ovulation prohormone after exercise (1.6 ± 0.1 versus 0.3 ± 0.1, normalized to two housekeeping genes, *p* < 0.01, Figure 4A).

It has been suggested that the serotonergic system is involved in the exercise effects [8,20] and oviposition in *L. stagnalis* [21]. We evaluated the expression of the gene encoding the rate-limiting enzyme in serotonin synthesis, tryptophan hydroxylase (TH), under the same conditions. The number of TH transcripts was significantly higher in the CNS of the exercised snails (2.9 ± 0.2, 1.6 ± 0.14, *p* < 0.05, Figure 4B).

### 3.3. Caudo-Dorsal Cells Producing Egg-Laying Hormones Are More Responsive to Electrical Stimulation in Snails Exercised in Shallow Water

The effect is seen only in the left CDC cluster.

The electrical coupling of CDC neuronal clusters suggests the synchronous operation of neurons located in the left and right cerebral ganglion. However, the clusters of neurons responsible for male sexual behavior, and that have inhibitory influence on CDC neurons, are asymmetrically located in the right cerebral and pedal ganglia (Figure 5) [24]. Taking this into account, we analysed the left and right CDC clusters separately.

There were no significant differences in the levels of MP in CDCs or in their initial electrical activity (neurons were silent in most cases). The mean level of MP was −59.6 mV in the left CDCs and −58.4 mV in the right CDCs in the control snails; in the exercised snails, these were −62.3 and −62.2 mV, respectively.

Significant differences were found in CDCs response to intracellular electrical stimulation of the left ganglion compared with the control (pairwise Wilcoxon test, number of pairs *n* = 7, *z* = 2, *p* < 0.05, Figure 6A,C). The number of spikes upon current injection (1 nA) was 1–4 in the control (7 preparations, 8 neurons) and up to 30 in the exercised group (7 preparations, 8 neurons). The median was 1 in the control group and 11 in the experimental group. Test for equal medians (Chi2) was 4.7 (*p* = 0.03).

There were no significant differences in MP levels, spiking activity or response to current injection in the right CDC cluster between the exercised snails (8 preparations, 8 neurons) and control snails (8 preparations, 11 neurons). The median was 1 in the control group and 0.5 in the experimental group (*z* = 1.26, *p* = 0.21, Figure 6B).

Therefore, we found that in exercised snails, the responsiveness of the left CDC neurons to excitatory stimulus was enhanced even before the snails were placed in clean water. 

### 3.4. Serotonin Has No Direct Effect on CDC Neurons

Exercise enhanced expression of the tryptophan hydroxylase (TH) gene involved in serotonin synthesis. The immediate serotonin precursor, 5-HTP—the product of TH—has been reported to enhance oviposition [21]. Thus, it seemed likely that enhanced release of serotonin is responsible for the observed changes in the reproductive behaviour caused by intense locomotion. We tested whether serotonin has a direct effect on the activity of CDC neurons. A total of four preparations (two controls and two exercised) and five different CDC neurons were used in these experiments. We also used the cerebral giant cell (CGC or C1, *n* = 2) from the cerebral ganglia as a control to test the effect of serotonin, as C1 is known to respond with excitation to serotonin [35]. We measured cellular effects of serotonin application immediately, after 15 min and after 90 min from the onset of application in a flow. Serotonin (0.05 mM) produced no changes in MP or spiking activity in all CDC neurons tested (Figure 7). In contrast, both C1 neurons responded to serotonin with depolarization and an increase in the spiking rate under the same conditions (Figure 7). Thus, we conclude that serotonin has no direct effect on the activity of CDC neurons.

## 4. Discussion

The aim of our study was to test the hypothesis that energy-consuming intense locomotion may stimulate reproductive behaviour in the model organism, *L. stagnalis*. Indeed, the number of egg clutches and the total number of eggs were significantly higher in exercised snails, while the number of eggs per clutch did not differ between the control and exercised snails. The observed increase in their fertility was thus mainly due to the activation of the oviposition behaviour and presumes central nervous regulation.

Under natural conditions, pond snails travel in shallow water or even on dry land when their water reservoir is drying up or when they need to leave it due to other unfavourable conditions. In these instances, they switch to intense terrestrial locomotion, which exists in their behavioural repertoire but is rarely required when they are fully submerged in water. The sequence of conditions that snails encountered in our experiments, namely home aquarium with “dirty” water, arena with shallow water and a new aquarium with clean water, may simulate the natural situation that occurs when a snail moves from a water reservoir they are accustomed to onto a new reservoir. We found that fertility in animals in these experiments increased compared with the control snails that were allowed to drift in water or to use underwater ciliary locomotion instead of intense crawling.

Activation of energy-consuming reproductive behavior following a period of intense locomotion, which also requires high energy expenditure, should have an adaptive significance. Moreover, the expected biological benefits must justify the energy expenditure. As a hypothesis, we proposed that intense locomotion is considered by the organism as a precursor to environmental changes. Augmented progeny production is an important means of preadaptation to novelty as it leads to increased diversity and better survival of genetic material.

All changes in lifestyle, including forced changes in locomotor behavior, could be considered potential stressors. However, the influence of intense locomotion in *Lymnaea* differs from other types of presumably stressful influences. We previously tested the effect of aquatic turbulence and found that it causes opposite changes in orienting behavior compared with crawling in shallow water [8]. Other “stressors”, such as starvation [36,37] or exposure to bright light at night [38] suppressed oviposition. Therefore, we cannot explain the observed effect using only a common stress-reaction.

Exercise-induced activation of reproductive behavior seems to depend on the season. The effect was more evident during the period of lengthening daylight hours, from January to May, in comparison to the September–December period. Snails were kept under a natural light cycle in our experiments; therefore, it is not surprising that this difference was observed. The light-cycle-dependent seasonal differences in reproduction and in the expression of ovulation prohormone was previously confirmed in *L. stagnalis* [39]. Interestingly, seasonal differences in reproduction and in monoamine metabolism were observed even under artificial uniform light cycle during the year [40]. Finding a mechanism that controls these season-dependent effects is an interesting avenue for future investigations. However, the biological reason for this phenomenon seems more or less clear. The chances of survival of parents and ancestors are higher during warm periods of the year, and increasing daylight is a reliable indicator of a “warmer future”. Since both intense locomotion and reproduction are associated with high energy costs, it makes no sense for survival to invest additional resources into reproduction during unfavourable conditions. It is known, for instance, that oviposition stops when snails are placed in low temperature (4–8 degrees) conditions [41].

As mentioned above, egg-laying is controlled by the ovulation prohormone [23,24]. Indeed, we detected an increase in the number of transcripts of ovulation prohormone in the CNS two hours after the end of exercise. The expression of tryptophan hydroxylase gene was similarly elevated. Thus, these RT-PCR data agree with the behavioural effects of exercise on egg-laying and with previous reports suggesting the involvement of serotonin in locomotion-induced neuronal and behavioural changes [8,20].

The CDC neurons, which produce the ovulation prohormone, switch to electrical activity prior to egg laying [28]. In most cases in our experiments, as in earlier studies, they were silent [42]. One explanation for the absence of spiking activity is that all experiments were performed immediately after exercise, while the number of egg clutches was evaluated 24 h after exercise. However, in a group of eight exercised snails residing in a single aquarium, we observed only up to four egg clutches, which means that not all snails responded to exercise by laying eggs. Hence, electrical activity of CDC neurons, even if tested within the proper period, is expected in less than half of individuals.

Nevertheless, we found differences in the biophysical characteristics of CDCs between the control and exercised snails even prior to the influence of clean water, which normally stimulates oviposition. After exercise, CDCs were more responsive to electrical stimulation (Figure 6). This increase in responsiveness can contribute to the mechanisms underlying the effect of exercise on egg-laying. Earlier, starvation, which suppresses egg-laying in behavioural experiments, was reported to cause hyperpolarization of the CDCs, thus elevating their spiking threshold [37] and inhibiting their discharge. Our results suggest that the excitability of these neurons, not manifesting itself in the shift of the resting MP, may be another possible mechanism of their functional regulation.

For the first time, to our knowledge, we observed functional asymmetry, i.e., the functional differences in symmetrically located and electrically connected clusters of neurons, after exercise. There were no significant differences between left CDCs and right CDCs in control snails. However, in exercised snails, an increase in responsiveness to electrical stimulation was observed in the left CDC cluster, but not in the right cluster. We speculate that this effect may be due to the asymmetrically located male hormonal system, which has a suppressive influence on the neurons associated with female-specific behaviour.

Several groups of neurons regulating male behavior form specific anatomic structures and are located only in the right part of the CNS, particularly the neurons of the anterior and ventral lobes of the right cerebral ganglion, pedal lateral lobe in the right pedal ganglion and some neurons in the right pleural and parietal ganglia, which innervate the penis ([24], Figure 5). It is known that some external factors have reciprocal influences on male and female reproductive behaviour. For example, a high density of snails in a limited aquatic space stimulates male behaviour and suppresses oviposition, while a lower density of snails has the opposite effect [24]. A number of substances released during the performance of the male behaviour or during egg laying may be involved in the regulation of the opposite sexual behaviour. For example, the APGW peptide, which activates male copulatory behaviour, hyperpolarizes the CDCs [43]. The FMRF peptide, which is synthesized in neurons associated with male behaviour, suppresses the spiking activity of the CDCs [44] and delays the initiation of a new period of their activity [42,43]. The seminal fluid peptide, ovipostatin, diminishes egg laying in the female-recipient mollusc [24].

Interestingly, the unpaired ring neuron, which is excited during stimulation of the penial nerve [45,46] and has inhibitory influence on CDCs, is located in the right cerebral ganglion [46]. This neuron is considered the best candidate for a central “switcher” between male and female behaviours. The ring neuron contains the APGW peptide [47], which, as already mentioned, hyperpolarizes the CGCs. Taking into account that extrasynaptic or volume release of neurotransmitters has been demonstrated in many parts of the *Lymnaea* CNS [20,30,48,49,50,51,52] and seems to be a very common phenomenon, it is likely that the release of APGW and other substances associated with male behaviour can have stronger inhibitory effect on the right CDC cluster than on the left CDC cluster.

Another well-known example of chemical asymmetry in the CNS of *Lymnaea* is presented by the two giant monoaminergic neurons located in the left and right pedal ganglia. One of these neurons, LPeD1, is serotonergic while the other, RPeD1, is dopaminergic. Serotonin and dopamine were reported to have opposite effects on oviposition in behavioural experiments [40]. The CDC activity can be inhibited by dopaminergic input [53,54]. Remarkably, the dopamine-secreting neuron, RPeD1, is located on the right, “male” side of the CNS. This asymmetry could also contribute to the observed difference between the response of the left and right CDCs to exercise. RPeD1 and LPeD1 have no direct projections to the cerebral ganglia; however, they can modulate multiple projections, from pedal to cerebral ganglia, differently, and thus may have some indirect effects on CDCs.

Serotonin is not only a neurotransmitter in the nervous system, it is also a neurohumoral factor, the role of which in reproductive behaviour and embryogenesis is being actively studied. Serotonin modulates the maturation of oocytes and sperm in the reproductive system of vertebrates and invertebrates [55,56,57]. Serotonin is found in oocytes of invertebrates [58,59,60,61] and vertebrates [62,63,64]. In *L. stagnalis*, an increase in the intracellular content of serotonin in the early stages of embryogenesis causes morphological and behavioural changes in young individuals [21,40]. In the same species, activation of serotonin synthesis increases fertility and influences the behaviour of offspring, increasing their locomotor activity [21,40]. Since activation of the serotonin system accompanies intense locomotion [8,20], it seems likely that serotonin can play a role in the mechanism responsible for the effect of intense locomotion on reproductive behaviour. However, we found no evidence of a direct effect of serotonin on the spiking activity or the MP of CDC neurons. CDCs are activated by neurons of the lateral lobes (LL) of cerebral ganglia [24]. The LL neurons seem to orchestrate both the development (via activation of neurons of the dorsal bodies) and the activity of the female reproductive system (via their influence on the CDC neurons). Hypothetically, serotonin may act at this level of hormonal regulation upstream of CDCs, indirectly activating the CDC neurons.

## 5. Conclusions

Our data suggest that intense crawling in shallow water enhances oviposition in *L. stagnalis.* The effect depends on the season, correlates with increased excitability of the CDC neurons and increases expression of the ovulation prohormone gene. The findings are in agreement with the hypothesis that intense locomotion activates biological mechanisms underlying preadaptation to possible environmental changes.

## Figures and Tables

**Figure 1 biology-12-00764-f001:**
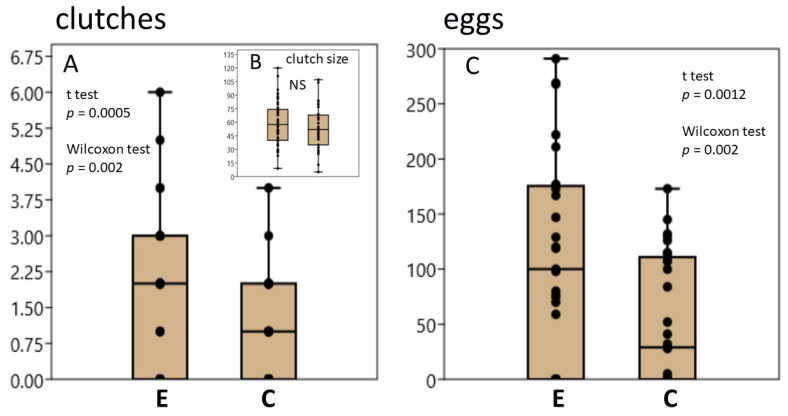
Crawling in shallow water stimulates oviposition in *L. stagnalis*. (**A**), Number of egg clutches in the exercised (E) and control (C) groups. Box and jitter plot (median, first and third quartiles, minimum and maximum). Sign test: *r* = 14, *p* = 0.0009; Monte Carlo: *p* = 0.0012; Exact test: *p* = 0.001. Results of *t*-test and paired Wilcoxon test are presented in the Figure. (**B**), Size of egg clutches (number of eggs per clutch) in the exercised (E) and control (C) groups. Tests for normal distribution: Shapiro–Wilk, W = 0.98 and 0.98, *p* (normal) = 0.46 and 0.82; Anderson–Darling, A = 0.41 and 0.24, *p* (normal) = 0.33 and 0.75. Tests for equal means *t* = 1.15, *p* = 0.25; Monte Carlo permutation: *p* = 0.25. (**C**), Number of eggs in the exercised (E) and control (C) groups. *t*-test *t* = 3.61, *p* (same mean) = 0.001; Sign test *r* = 17, *p* = 0.017; Wilcoxon test: *z* = 3.036, *p* = 0.002; Monte Carlo *p* = 0.001; Exact: *p* = 0.001. The data for A, B and C (median, first and third quartiles, min and max) are presented using box and jitter plots.

**Figure 2 biology-12-00764-f002:**
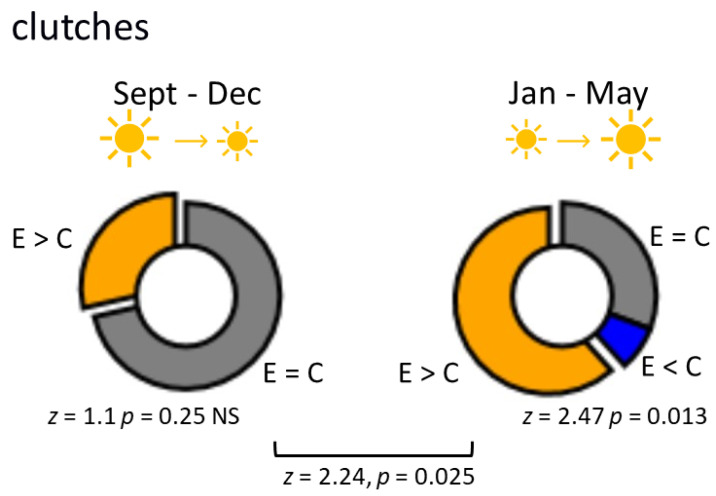
The effect of intense locomotion on egg-laying behaviour is season-dependent. Percentage of experiments in which the number of clutches were higher in the experimental group (yellow); in which the number of clutches were equal in the experimental (E) and control (C) groups (grey); in which the number of clutches were lower in the experimental group than in the control (blue) in the September–December (left, *n* = 14)) and January–May (right, *n* = 15)) periods. Pairwise Wilcoxon test suggests no differences during the September–December period and significant differences during the January–May period. The difference between the results obtained in the two periods was confirmed using Fisher’s exact test, which was used to evaluate season-dependent differences in the enhancing effect of exercise on oviposition.

**Figure 3 biology-12-00764-f003:**
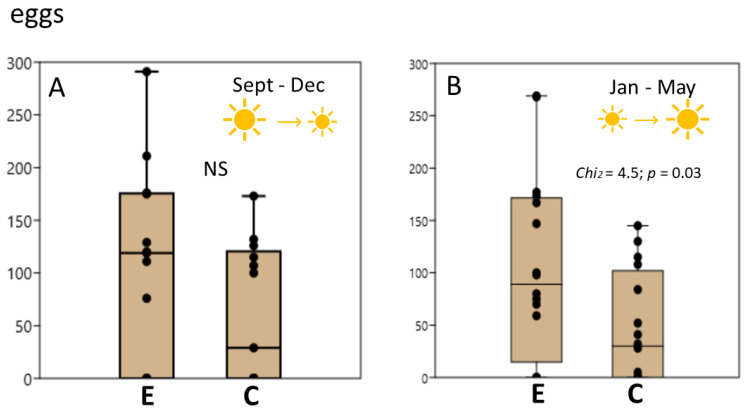
The number of eggs laid after exercise in different seasons. (**A**), The difference in the number of eggs between the control (C) and exercised snails (E) is not significant when evaluated during decreasing daylight hours in September–December. (**B**), In the January–May period, the difference in the number of eggs between control and exercised snails is significant.

**Figure 4 biology-12-00764-f004:**
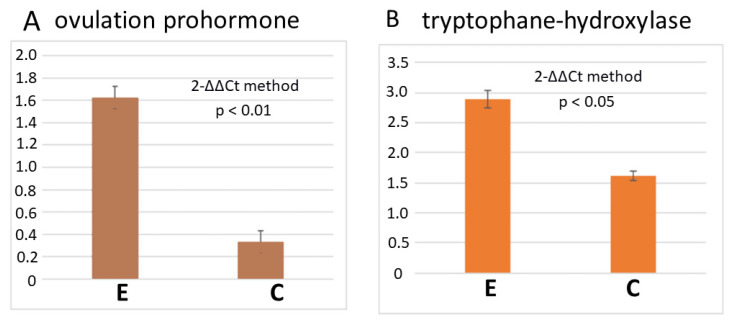
Changes in the expression of ovulation prohormone and tryptophan hydroxylase genes after intense crawling and resting in water. (**A**) The level of expression of ovulation prohormone in exercised and control snails. Exercised snails were rested in water for 2 h after exercise (E); the control snails (C) were kept in water containers, handled and placed in a new container of water for 2 h. (**B**) The level of expression of the tryptophan hydroxylase gene in exercised and control snails. Exercised snails were rested in water for 2 h after exercise; the control snails were kept in water containers, handled and placed in a new container of water for 2 h.

**Figure 5 biology-12-00764-f005:**
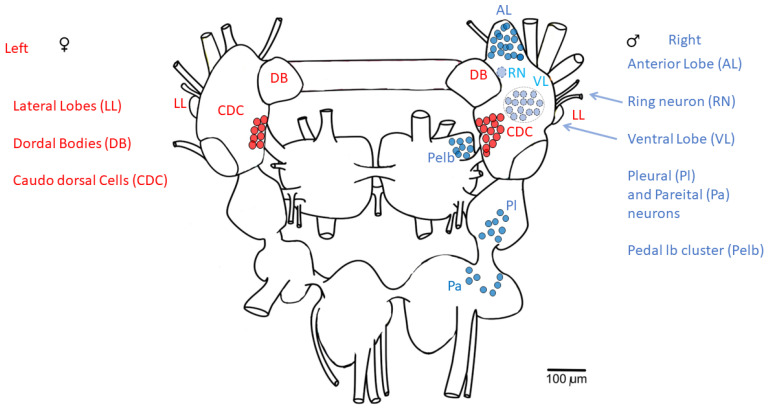
Asymmetric location of neurons associated with male and female behaviours in the CNS of *L. stagnalis*. Schematic representation of the central ganglia of *L. stagnalis*, dorsal view (modified after [25]) showing the neurons associated with male copulatory and female egg-laying behaviours (modified after [24]). The male system is depicted in blue, the female in red. The light-blue colour indicates the ventral position of neurons. Several groups of neurons regulating male behavior form specific anatomic structures (clusters) and are located only in the right part of the CNS: anterior and ventral lobes of the right cerebral ganglion and pedal lateral lobe in the right pedal ganglion. The unpaired ring neuron (RN), which has inhibitory influence on CDCs via the APGW peptide, is located in the right cerebral ganglion.

**Figure 6 biology-12-00764-f006:**
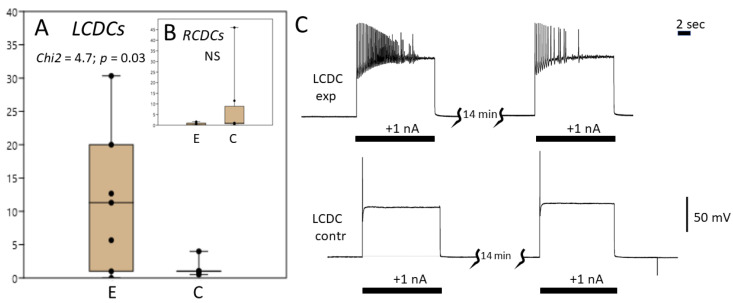
Intracellular recording from the caudo-dorsal neurons in control and exercised snails. (**A**) Number of spikes in the left CDC neurons in response to injection of 1 nA current for 10 s. E: exercised (7 preparations, 8 neurons); C: control (7 preparations, 8 neurons); Box and jitter plot (median, first and third quartiles, minimum and maximum). Left CDCs demonstrate enhanced excitatory effect to stimulation after exercise. Tests for equal medians: Chi2 = 4.7, *p* = 0.03. Paired Wilcoxon test: *z* = 2.03, *p* = 0.04. (**B**) Number of spikes in the right CDC neurons in response to injection of 1 nA current for 10 s. E: exercised (8 preparations, 8 neurons); C: control (8 preparations, 11 neurons); Box and jitter plot (median, first and third quartiles, minimum and maximum). No difference was observed between the experimental and control preparations. Tests for equal medians: Chi2 = 0.41026, *p* = 0.52; Paired Wilcoxon test: *z* = 1.26, *p* = 0.21. (**C**) Records of electrical activity in the left CDC neurons in response to stimulation (1 nA, 10 s). Upper trace: exercised snails (MP −78 mV); lower trace: control snails (MP −62 mV).

**Figure 7 biology-12-00764-f007:**
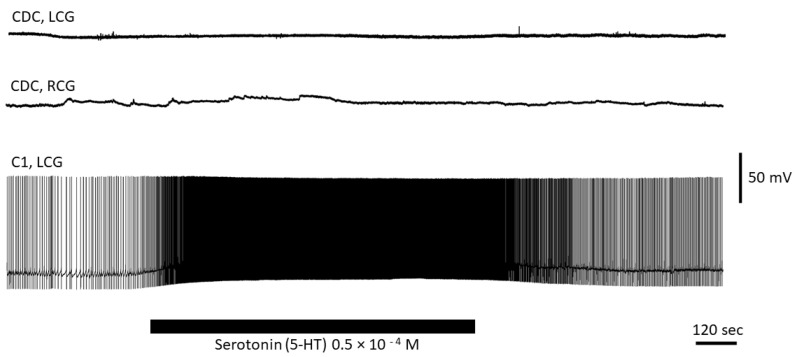
Effects of serotonin on the electrical activity of CDC and CGC neurons. Serotonin has no direct effect on electrical activity of CDCs (upper traces) but excites the C1 (giant serotonergic neuron) in the same experiment (lower trace). LCG: left cerebral ganglion; RCG: right cerebral ganglion.

## Data Availability

All data that support the findings of this study are available from the corresponding authors upon reasonable request.

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
