# Peer review of "Intense Locomotion Enhances Oviposition in the Freshwater Mollusc Lymnaea stagnalis: Cellular and Molecular Correlates"

_biology, 2023, doi:10.3390/biology12060764_

Round 1

Reviewer 1 Report

In this study, Chistopolsky et al., investigated the hypothesis that in the pond snail Lymnaea stagnalis intense locomotion (i.e., crawling in shallow water) would stimulate oviposition in a novel environment. The results of the experiment performed confirmed this hypothesis, suggesting that 2-hour crawling in low water can stimulate oviposition but the effects is seasonally-dependant. Moreover, the data presented suggest that this behavioral response may result from the upregulation of the expression levels of the egg-laying prohormone gene CDCH-1.

GENERAL COMMENTS

·        The topic of the paper well fits within the scope of Biology.

·        The title properly reflects the subject of the paper

·        The abstract provides an accessible summary of the manuscript, but it should be shortened

·        The paper’s premise is interesting and important.

·        The keywords accurately reflect the content.

·        The introduction sets out the argument, summarizes recent research related to the topic, and highlights gaps in current understanding or conflicts in current knowledge.

·        The results are well discussed, and conclusions also consider the limits of the study.

·        The methods are appropriate, and the results are clearly presented.

·        The figures are quite clear, but I suggest to include a figure summarizing the study design

·        The paper has an appropriate length.

·        References are balanced, updated, and quite complete.

I would consider the manuscript for publication only after the following major revisions are performed.

ABSTRACT:

·        I suggest to provide a shorter and less speculative abstract

·        Lymnaea stagnalis should be italicized (e.g., Lines 17, 28)

·        I suggest to include a graphical abstract: the data are interesting and could be useful for the naïve readers

INTRODUCTION:

-        I suggest including a short paragraph indicating the hypothesis of the study and the rationale of the study

-        Line 60: I suggest including the review by Rivi et al., (2021) which summarizes the validity of Lymnaea as a model for translational neuroscience

METHODS:

-        A study design paragraph could be helpful. Authors should divide the study in different experiments. This may be helpful for the readers

-        What kit was used for RNA extraction and retrotranscription? What parameters were used?

-        Please, indicate some studies in which EF1alpha and GAPDH were used as the housekeeping genes

-        Statistics analysis is missing . Authors should indicate all the statistical analyses performed, as well as the software used for the analysis in the method section. I’ve seen them in the figures and figure legends, but I think it is important to provide a specific section in the Methods

-        Where the data checked for normal distribution? If so, how?

RESULTS

-        Fig. 2 Legends. It should be indicated that E = experimental group and C = control snails

Minor errors

Author Response

Thank you very much for your comments and suggestions, they are very helpful! We have addressed all of them, and our response to each comment is in italic font for your convenience.

ABSTRACT:

  • I suggest to provide a shorter and less speculative abstract

Done, we have omitted the following sentences: “It was hypothesized that increased motor activity suggests challenges ahead, in anticipation of which the above adjustments are made. Enhanced reproduction seems to be an important means of adaptation to novel environments as it serves to increase the number and variety of descendants.”

  • Lymnaea stagnalis should be italicized (e.g., Lines 17, 28)

Thank you for the comment. L. stagnalis is italicized

  • I suggest to include a graphical abstract: the data are interesting and could be useful for the naïve readers

It had been provided separately, we have now inserted the graphical abstract into the main document

INTRODUCTION:

-        I suggest including a short paragraph indicating the hypothesis of the study and the rationale of the study

The para 3 contains now both the hypothesis and rationale. We have added the following sentence to make it more clear. “These data suggesting an activation of serotonin system by intense crawling and stimulating effect of serotonin on oviposition became an additional rationale for our hypothesis on stimulating effect of intense locomotion on the reproductive behavior in spite of obvious energetic costs of both exercise and reproduction.”

-        Line 60: I suggest including the review by Rivi et al., (2021) which summarizes the validity of Lymnaea as a model for translational neuroscience

Thank you, done, we have included both reviews Rivi et al. 2020 and 2021. Both are highly related to the statement 

METHODS:

-        A study design paragraph could be helpful. Authors should divide the study in different experiments. This may be helpful for the readers

Yes, it is a good idea, we have added this para 2.1.

-        What kit was used for RNA extraction and retrotranscription? What parameters were used?

We have now added the missing information about these methods.

-        Please, indicate some studies in which EF1alpha and GAPDH were used as the housekeeping genes

Thank you, we relied on the following work, which is now cited. Young AP, Landry CF, Jackson DJ, Wyeth RC. Tissue-specific evaluation of suitable reference genes for RT-qPCR in the pond snail, Lymnaea stagnalis. PeerJ. 2019 Oct 15;7:e7888. doi: 10.7717/peerj.7888.

-        Statistics analysis is missing. Authors should indicate all the statistical analyses performed, as well as the software used for the analysis in the method section. I’ve seen them in the figures and figure legends, but I think it is important to provide a specific section in the Methods

In the first version, we mentioned the analysis that was used in each experiment at the end of each method description. Now, we formed an additional section Statistics, in which all stat methods used in different experiments are listed and justified.

-        Where the data checked for normal distribution? If so, how?

Yes, the analysis for normal distribution was done. It is now stated in the Methods

RESULTS

-        Fig. 2 Legends. It should be indicated that E = experimental group and C = control snails

Thank you. Done

Reviewer 2 Report

In this study, the authors investigated the mechanisms underlying the increased oviposition caused by intense locomotion in the pond snail Lymnaea stagnalis. Although the findings are of potential interest, some concerns must be addressed by the authors before the manuscript can be considered for publication.

 Major comments

1) Overall, the authors must provide further description and substantiation for the statistical tests that were used in the study. The lack of this information causes confusion about what test was used for the different types of experiments, behavioral, molecular, and electrophysiological. Limited details were outlined in the Material and Methods. Indeed, tests, such as Shapiro-Wilk, Anderson-Darling, Monte Carlo permutation, Chi square, were only mentioned briefly in some of the figure legends without any rationale of why a given test was utilized. In this regard, for example, it is completely unclear how the data shown in Fig. 2 were analyzed.

2) Regarding the analysis of the electrophysiology data, it is unclear why the authors decided to utilize the Pairwise Wilcoxon test. If the comparison was conducted in neurons from exercised and control animals, the statistical test used should not be the Pairwise Wilcoxon test, which compares data in a before-after design. Authors should have instead used the Mann-Whitney tests which is designed to compared different groups, like control and exercised.

3) In the Results, the authors state that “The above behavioral finding suggested that exercise potentiates the stimulating effect of clean water on oviposition” (lines 221-222). The statement confuses the outcome from the behavioral experiments, and the main claim of the study overall. Does intense locomotion stimulate oviposition, or does it enhance oviposition that is normally induced when animals are moved into clean water? The authors should carefully address this discrepancy.

4) It is not clear why the expression of tryptophan hydroxylase was investigated. Given that tryptophan hydroxylase is the first the enzyme in the synthesis of 5-HT, one would think that the authors selected its expression to analyze changes in 5-HT synthesis caused by exercise. However, this link was not made explicit in text.

5) The two house-keeping genes, which were used to normalize the molecular data (line 227), were not listed and the rationale for their utilization was not provided.

6) Regarding the electrophysiology data, it is not clear why neuronal excitability was measured twice. In addition, the rationale for waiting 14 minutes between excitability measurements must be provided.

7) In the Discussion, the authors commented on the possible role of intense locomotion as a potential stressor (lines 335-337). If intense locomotion functions as a stressor, it seems counterintuitive that the snails would lay eggs at the end of an event that might be perceived as dangerous for the offspring. The authors should clarify this point.

Minor comments

1) In the Abstract (line 17), the term “Lymnaea stagnalis” must be italicized.

2) In the Abstract (line 24), the term “stimulation” must be better defined.

3) Throughout the manuscript, the term “tryptophane” must be replaced with “tryptophan”.

Please check the manuscript for punctuation and typos.

Author Response

Many thanks for your helpful comments! We have addressed all of them, and our response to each comment is in italic font for your convenience.

 Major comments

  • Overall, the authors must provide further description and substantiation for the statistical tests that were used in the study. The lack of this information causes confusion about what test was used for the different types of experiments, behavioral, molecular, and electrophysiological. Limited details were outlined in the Material and Methods. Indeed, tests, such as Shapiro-Wilk, Anderson-Darling, Monte Carlo permutation, Chi square, were only mentioned briefly in some of the figure legends without any rationale of why a given test was utilized. In this regard, for example, it is completely unclear how the data shown in Fig. 2 were analyzed.

Thank you for this comment. We now insert the para considering Statistics used in each experiment into the Methods. We also explain in the Figure 2 legend how statistics were calculated:

“Pairwise Wilcoxon test suggests no difference during September-December period and significant difference during January to May period. The difference between the results obtained in the two periods is confirmed by Fisher's exact test that was used to evaluate the season-dependent differences in the enhancing effect of exercise on oviposition”.

  • Regarding the analysis of the electrophysiology data, it is unclear why the authors decided to utilize the Pairwise Wilcoxon test. If the comparison was conducted in neurons from exercised and control animals, the statistical test used should not be the Pairwise Wilcoxon test, which compares data in a before-after design. Authors should have instead used the Mann-Whitney tests which is designed to compared different groups, like control and exercised.

Since in each experiment we had a pair of control and experimental preparations, placed in the same dish and treated simultaneously, pairwise Wilcoxon test was used to evaluate the difference in activity, the membrane potential level and number of spikes in response to 1nA current injection between the experimental and control preparations. If in one preparation more than one neuron was investigated the average for this preparation was used for pairwise statistic analysis. In addition, the data was evaluated by tests for independent groups for equal medians Chi2.

  • In the Results, the authors state that “The above behavioral finding suggested that exercise potentiates the stimulating effect of clean water on oviposition” (lines 221-222). The statement confuses the outcome from the behavioral experiments, and the main claim of the study overall. Does intense locomotion stimulate oviposition, or does it enhance oviposition that is normally induced when animals are moved into clean water? The authors should carefully address this discrepancy.

---- Indeed, although we observed changes in the state of CDC neurons and gene expression after exercise only, all behavioral experiments were performed with additional stimulation with clean water (both control and exercised snails). We changed the word “stimulate” for “enhance” as you suggested.

  • It is not clear why the expression of tryptophan hydroxylase was investigated. Given that tryptophan hydroxylase is the first the enzyme in the synthesis of 5-HT, one would think that the authors selected its expression to analyze changes in 5-HT synthesis caused by exercise. However, this link was not made explicit in text.

--We add this info now into the abstract, introduction and methods, see 2.1. Study design

5) The two house-keeping genes, which were used to normalize the molecular data (line 227), were not listed and the rationale for their utilization was not provided.

- corrected.

6) Regarding the electrophysiology data, it is not clear why neuronal excitability was measured twice. In addition, the rationale for waiting 14 minutes between excitability measurements must be provided.

The neuron was stimulated several times with an interval of 10 -14 min to check the stability of the response. The average value was taken into further analysis.

7) In the Discussion, the authors commented on the possible role of intense locomotion as a potential stressor (lines 335-337). If intense locomotion functions as a stressor, it seems counterintuitive that the snails would lay eggs at the end of an event that might be perceived as dangerous for the offspring. The authors should clarify this point.

We have rewritten this para as follows to clarify it:

All changes in the lifestyle, including forced changes in the locomotor behavior, could be considered as potential stressors. However, the influence of intense locomotion differs from other types of presumably stressful influences in Lymnaea. Previously, we tested the effect of an aquatic turbulence, and found that it causes opposite changes in the orienting behavior compared to crawling in low water (Aonuma et al., 2020).  Other “stressors”, such as starvation (Dogterom et al., 1984b; Ter Maat et al., 1982) or exposure to bright night light (Baz et al., 2022) suppressed oviposition. Therefore, we can not explain the observed effect just by a common stress-reaction.

Minor comments

In the Abstract (line 17), the term “Lymnaea stagnalis” must be italicized.

  • L. stagnalis is italicized

2) In the Abstract (line 24), the term “stimulation” must be better defined.

We changed it to “enhance”.

3) Throughout the manuscript, the term “tryptophane” must be replaced with “tryptophan”.

Thank you for the correction, done.

Comments on the Quality of English Language

Please check the manuscript for punctuation and typos.

We did our best!

Round 2

Reviewer 2 Report

The authors have satisfactorily addressed the comments that were raised